# Ten minutes of transcranial static magnetic field stimulation does not reliably modulate motor cortex excitability

Sabrina Lorenz*, Birte Alex, Thomas Kammer

Department of Psychiatry, Section for Neurostimulation, University of Ulm, Ulm, Germany

* sabrina.lorenz@uni-ulm.de

## Abstract

Recently, modulatory effects of static magnetic field stimulation (tSMS) on excitability of the motor cortex have been reported. In our previous study we failed to replicate these results. It was suggested that the lack of modulatory effects was due to the use of an auditory oddball task in our study. Thus, we aimed to evaluate the role of an oddball task on the effects of tSMS on motor cortex excitability. In a within-subject-design we compared 10 minutes tSMS with and without oddball task. In one of the two sessions subjects had to solve an auditory oddball task during the exposure to the magnet, whereas there was no task during exposure in the other session. Motor cortex excitability was measured before and after tSMS. No modulation was observed in any condition. However, when data were pooled regarding the order of the sessions, a trend for an increase of excitability was observed in the first session compared to the second session. We now can rule out that the auditory oddball task destroys tSMS effects, as postulated. Our results rather suggest that fluctuations in the amplitudes of single pulse motor evoked potentials may possibly mask weak modulatory effects but may also lead to false positive results if the number of subjects in a study is too low. In addition, there might be a habituation effect to the whole procedure, resulting in less variability when subjects underwent the same experiment twice.

## Introduction

Over the past years, several studies have demonstrated that the application of a focal static magnetic field (transcranial static magnetic field stimulation, tSMS) can modulate the excitability of the targeted brain area [e.g., 1, 2–6]. Initially, this method was investigated in the motor system [1]. In their study, a strong permanent magnet was held over the motor region at subject's scalp for 10 minutes. Reflecting the instantaneous excitability of the motor system [7], the amplitude of motor evoked potentials (MEP) evoked by single pulses of transcranial magnetic stimulation (TMS) was measured before and after tSMS. The result was a decrease in MEPs for 6 minutes after tSMS, indicating reduced excitability of the stimulated cortical area [1]. This effect has been replicated in another study, applying tSMS for 15 minutes [2]. An effect of tSMS on cortical excitability comparable to those evoked by other non-invasive brain

**Competing interests:** The authors have declared that no competing interests exist.

stimulation methods such as repetitive TMS [8–10] or transcranial direct current stimulation [11–13], but without the side effects [14, 15] would offer a wide range of possibilities for, i.e., treatment of neurological or psychiatric diseases. Therefore, our group decided to replicate the results of the initial study [1]. Unfortunately, we failed to observe a decrease in MEP amplitudes after tSMS application [16].

Subsequently, it was mentioned [17] that there was a difference between our study [16] and the former one [1], possibly accounting for the different results. To standardize subject's cognitive activity during magnetic exposure, we used an auditory oddball task. In contrast, there was no special task during exposure in other published tSMS studies.

It was suggested that the use of an oddball task could offer an explanation for the difference in results [17]. Counting mentally could activate the motor cortex due to individual finger counting habits [18], and these habits might manipulate the structure of mental number representations [19]. It has also been observed that productive and perceptive linguistic tasks increase motor cortex excitability bilaterally, indicating that forming number words mentally while counting could cause activation in the hand representation of the motor cortex [20]. Activation of the motor cortex without motor output might therefore interfere with neuroplastic modulation induced by static magnetic fields. In a reply to the letter [17], we discussed the possible effects of the oddball task on the excitability of the motor cortex and also mentioned its role in controlling subject's attention [21]. However, we opted for a direct experimental comparison as the best way to address this question.

Therefore, in the present study we aimed to test the influence of an acoustic oddball task on the effect of tSMS on motor cortex excitability by directly comparing static magnetic field stimulation with and without oddball task in a within-subject-design.

## Material and methods

### Subjects

Unfortunately, in the work of Oliviero et al., 2011 no effect size has been reported. Therefore, we based our power estimation (G*Power 3.1.7) on the effect size reported more recently [5]. In their analysis on MEP ratio for the left hand, they observed a significant Time by Group- interaction with an effect size of $\eta2 = 0.163$. Together with their sample size of n = 20 (between- group) and an assumed $\alpha$-error of 0.05, the power of their results was 0.24. Thus, using those values the calculated sample size for our experiment was 22. We decided to recruit 24 subjects to achieve appropriate power. In four cases the experiment could not be completed. Two subjects discontinued the experiment due to an uncomfortable feeling during the experiment. One session had to be cancelled due to MEP amplitudes being too low, even with high TMS intensities up to 66% of maximum stimulator output (MSO). One further session had to be cancelled due to technical difficulties. Therefore, the entire experiment was finished in 20 healthy subjects.

All subjects were right-handed according to a modified version of the Edinburgh Inventory Scale [22] and were screened for neurological and psychiatric disorders, chronic illnesses, previous head surgeries, metal implants in the head region and drug abuse. All subjects gave their written informed consent and were paid for their participation. The study followed the declaration of Helsinki and all experiments were approved by the Ethics Committee of the University of Ulm (231/17).

### Measurement of motor cortex excitability

MEPs of the right first dorsal interosseus muscle (FDI) at rest were evoked by monophasic single pulses of TMS using a Magstim 200 stimulator (Magstim Co., Whitland, UK), connected to a figure-of-eight-coil (Double 70mm Alpha Coil, Magstim Co., Whitland, UK).

To identify the motor „hotspot"of the right FDI, suprathreshold TMS pulses were applied to the subjects' left motor cortex. The coil was held tangentially to the scalp with the handle pointing backwards in an angle of 45˚ to the sagittal plane. The scalp site with the highest MEP amplitudes was defined as the motor hotspot. To maintain the correct coil position during the experiment and to retrieve the hotspot in the second session, we used a neuronavigation system (PowerMAG View!, MAG & more, Munich, Germany). The hotspot was additionally marked on the scalp using the pointer for correct positioning of the permanent magnet.

The resting motor threshold (RMT) was measured at the beginning of each session. RMT was defined as the TMS intensity generating an MEP of about 50 µV in at least 5 out of 10 trials.

MEPs were monitored using single TMS pulses. To minimize anticipation and habituation, the pulse frequency jittered randomly between 0.125 Hz and 0.2 Hz (inter-stimulus interval: 5–8 sec).

For probing cortico-spinal excitability, we adjusted stimulation intensity to evoke MEPs with a mean amplitude of about 1mV in the particular subject. 10 single pulses were applied with a given intensity above RMT. If the mean amplitude was not between 0.8 mV and 1.3 mV, the procedure was repeated with an adjusted stimulation intensity, otherwise the tested intensity was defined as 1-mV-intensity.

MEPs of the right FDI were recorded using surface electrodes in a belly-tendon montage. Signals were bandpassed (10–2000 Hz) and amplified using a Toennies universal amplifier (Erich Jaeger GmbH, Hochberg, Germany), sampled with 5000 Hz and online presented, analyzed, and stored on a PC for offline-analysis using DasyLab 13.0 (measX GmbH und Co. KG, Mönchengladbach, Germany).

An acoustic and visual feedback signal was used for online control of relaxation of the FDI during MEP recording.

## Static magnetic field stimulation

For tSMS, we used the same magnet as in previous tSMS experiments [1, 2, 16], a cylindrical neodymium magnet (NdFeB) of 30 mm height and 45 mm diameter (model S-45-30-N, Supermagnete, Gottmandingen, Germany). The magnet was held manually on the scalp centered over the previously identified motor hotspot and tSMS was applied for 10 minutes with the south pole pointing to the scalp.

## Experimental design

The experiment included two sessions for each subject in a randomized order, which were separated at least by one week. In one session, during tSMS subjects had to perform an acoustic oddball task, whereas in the other session tSMS was applied without any additional activity of the subject. In the oddball task, subjects heard a sequence of two beep tones with different frequencies (300 and 500 Hz, 0.2 s each, every 2.5 seconds) through in-ear-headphones, and were asked to silently count the rare beeps without using their fingers [cf. 16, 23]. The number of rare beeps was adjusted between 20 and 30 in a 10 minutes train. In both sessions, subjects were asked to keep their eyes open and remain calm and silent.

In each session, the subjects were seated in a comfortable chair. The motor hotspot was located as described above before RMT and 1-mV-threshold were determined. Baseline measurement of MEPs (pre tSMS) lasted for 4 minutes. TSMS was then applied for 10 minutes by an additional experimenter. Post measurement started 1 minute after tSMS treatment and lasted 10 minutes.

## Data analysis

Statistics were performed using Statistica (V.13, StatSoft GmbH, Hamburg, Germany). Data were visually inspected for spontaneous motor activity 800ms before and after every MEP. MEPs with obvious pre-innervation were excluded. If the excluded data of one subject was above 7% of the subject's total data, the subject was excluded entirely, leading to the exclusion of two subjects due to numerous pre-innervation. Thus, data of 18 subjects were used for the analysis. Peak-to-peak amplitudes of MEPs of each session were summarized in 2 pre- and 5 post time points, thus each time point reflects the mean amplitude of all MEPs of 2 minutes. Mauchly's sphericity test was applied for repeated-measure data and in case of violation Greenhouse-Geyser correction was used.

Raw data of the 18 subjects included in the analysis can be seen in S1 File.

# Results

18 subjects (8 male, mean age 22.78 ± 2.4 years) entered the final analysis.

## Main analysis

Mean RMT was 36.9 ± 5.3% of maximum stimulator output (MSO) in the sessions with and 37.1 ± 5.7%MSO in the sessions without oddball task. A paired t-test showed that there was no significant difference in RMT between the two sessions (t = 0.36, p = 0.73), but a high correlation as revealed by correlation analysis (r = 0.89, p<0.001).

Mean TMS intensity to generate MEP amplitudes of about 1 mV was 50.6 ± 9.4% MSO in sessions with and 51.6 ± 10.4% MSO in sessions without oddball task. There was no significant difference between the two sessions (t = 0.77, p = 0.45), but a high correlation (r = 0.87, p<0.001).

In sessions with oddball task, raw baseline MEP amplitude was 1.08 ± 0.49 mV at time point $pre_1$ and 1.13 ± 0.61 mV at time point $pre_2$. In sessions without oddball task, raw baseline MEP amplitude was 1.07 ± 0.39 mV at time point $pre_1$ and 1.11 ± 0.44 mV at time point $pre_2$. There was no difference in MEP amplitudes between the two baseline time points of each session (with oddball task: t = 0.87, p = 0.40; without oddball task: t = 0.58, p = 0.57). Thus, we averaged MEPs of the two baseline time points for each session as $pre_{mean}$ value. There was no significant difference in $pre_{mean}$ MEP amplitudes between sessions with and without oddball task (t = 0.07, p = 0.94). However, there was no correlation between $pre_{mean}$ MEP amplitudes of the two sessions (r = 0.03, p = 0.92).

MEP amplitudes were then normalized for each subject with respect to the $pre_{mean}$ MEP amplitude of each session, respectively.

Data were analyzed using a two-factor analysis of variance (ANOVA) with the within-factors SESSION (with, without oddball) and TIME (1–7). There was no significant effect for any factor (SESSION: $F_{(1,17)}$ = 0.04, p = 0.84; TIME: $F_{(6,102)}$ = 1.02, p = 0.42) and no interaction ($F_{(6,102)}$ = 0.27, p = 0.95). For graphical illustration of the main analysis see Fig 1.

## Analysis with respect to session order

To investigate whether the order of the sessions has an impact on our results, we analyzed the data again, sorted by session order.

Mean RMT was 36.5 ± 5.8% of maximum stimulator output (MSO) in the first sessions and 37.5 ± 5.2%MSO in the second sessions. A paired t-test showed that there was no significant difference in RMT between the two sessions (t = 1.73, p = 0.10), but a high correlation as revealed by a correlation analysis (r = 0.91, p<0.001).

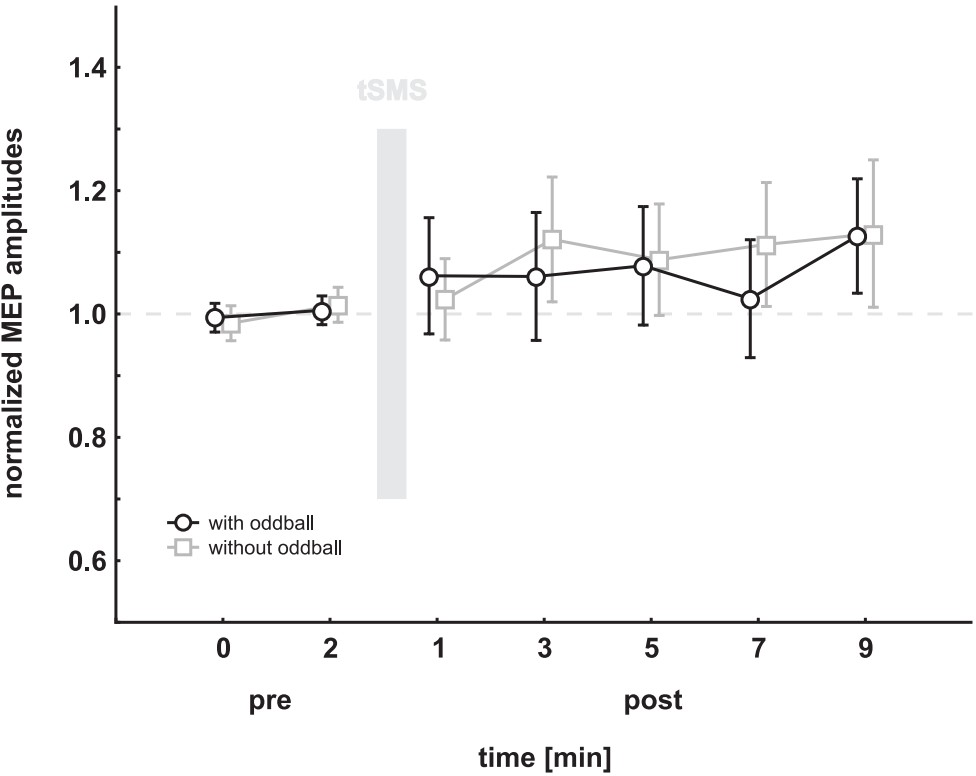

**Fig 1. Results of the experiment.** Mean normalized MEP amplitudes are shown pre and post magnet exposure in 2 min intervals. The grey bar represents the exposure (10min) to the permanent magnet. The post measurement started 1 minute after the end of magnet exposure. Error bars show standard error of the mean. No difference between the sessions with and without oddball task was observed, neither was any effect of magnet exposure.

Mean TMS intensity to generate MEP amplitudes of about 1 mV was 51.2 ± 10.7% MSO in the first sessions and 50.9 ± 9.1% MSO in the second sessions. There was no significant difference between the two sessions (t = 0.22, p = 0.83) but a high correlation (r = 0.87, p<0.001).

In the first sessions, raw baseline MEP amplitude was 1.04 ± 0.48 mV at time point $pre_1$ and 1.12 ± 0.63 mV at time point $pre_2$. In the second sessions, raw baseline MEP amplitude was 1.10 ± 0.40 mV at time point $pre_1$ and 1.12 ± 0.40 mV at time point $pre_2$.

There was no difference in MEP amplitudes between the two baseline time points of each session (first session: t = 1.04, p = 0.31; second session: t = 0.27, p = 0.79). Thus, we summarized the two baseline time points for each session as $pre_{mean}$ value. There was no significant difference in pre mean MEP amplitudes between the first and the second sessions (t = 0.15, p = 0.88). However, there was no correlation between $pre_{mean}$ MEP amplitudes of the two sessions (r = 0.03, p = 0.92).

Normalized MEP data were then analyzed using a two-factor ANOVA with the within-factors SESSION (first, second) and TIME (1–7). There was no significant effect for any factor (SESSION: $F_{(1,17)}$ = 3.90, p = 0.06; TIME: $F_{(6,102)}$ = 1.02, p = 0.42), but a trend towards significant differences between the two sessions as well as a significant SESSION*TIME interaction ($F_{(6,102)}$ = 2.47, p = 0.03, $\eta^2$ = 0.13). However, since sphericity of the data was violated as revealed by Mauchly's sphericity test, Greenhouse-Geyser correction was applied, resulting in a trend to a significant interaction only ($F_{(3.1,52.2)}$ = 2.47, p = 0.07). For graphical illustration

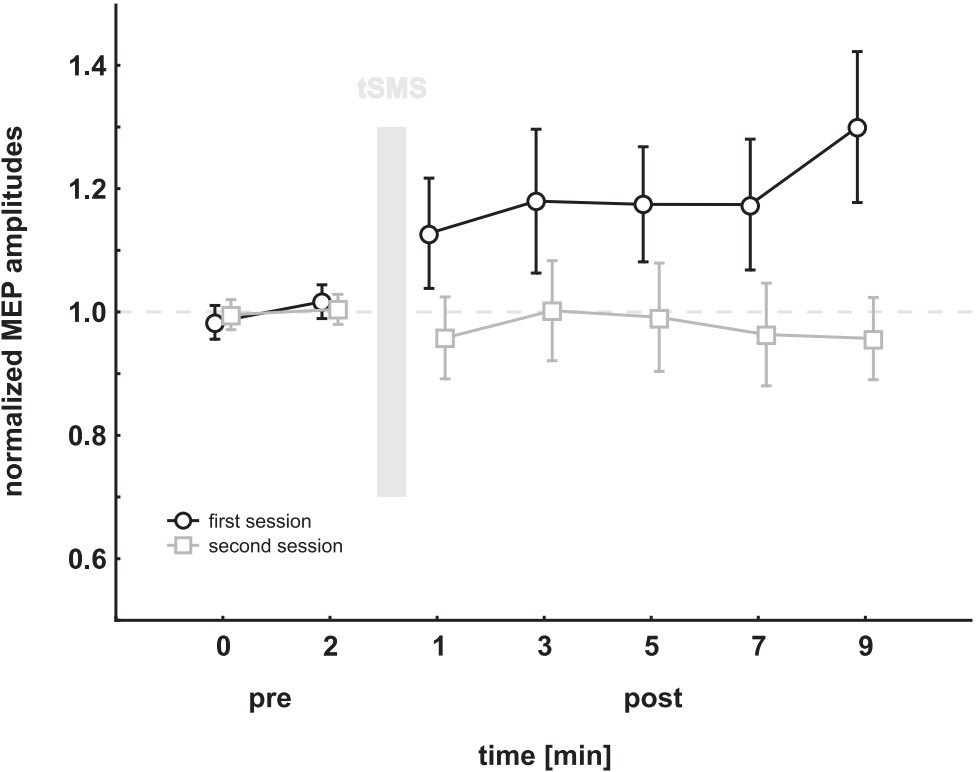

**Fig 2. Results of the experiment, grouped with respect to session order.** Mean normalized MEP amplitudes are shown pre and post magnet exposure in 2 min intervals. The grey bar represents the exposure (10min) to the permanent magnet. The post measurement started 1 minute after the end of magnet exposure. Error bars show standard error of the mean. There was a trend towards a significant difference between the first and the second session as well as a trend to increased MEP amplitudes after magnet exposure in the first session.

see Fig 2. Inspection of single subject data revealed, that the difference between session 1 and 2 was driven by only 4 out of the 18 subjects (see Fig 3).

Furthermore, a friendly anonymous reviewer encouraged us to report a 3-way ANOVA including the factors SESSION (first/second), INTERVENTION (with/without oddball), and TIME (1–7).

There was a significant effect for the factor SESSION ($F_{(1,224)}$ = 12.35, p<0.001, $\eta^2$ = 0.05). However, there was no other significant effect for any factor (INTERVENTION: $F_{(1,224)}$<0.001, p = 0.99; TIME: $F_{(6,224)}$ = 0.66, p = 0.68) nor any statistically significant interaction.

Please note that in this 3-way ANOVA the group of 18 subjects is splitted so that only 9 subjects are averaged for the factors SESSION and INTERVENTION. Furthermore, since our aim was to investigate the influence of the oddball-task on MEP Amplitude (i.e. INTERVENTION*TIME), the question of the influence of session order is a post-hoc question.

## Discussion

We evaluated the influence of an acoustic oddball task on the effect of tSMS on motor cortex excitability by directly comparing magnet exposure with and without oddball task. We could not find significant differences for any of the two conditions, indicating that the oddball task was not the reason for our failure to replicate tSMS effects [16], as postulated [17]. Since we

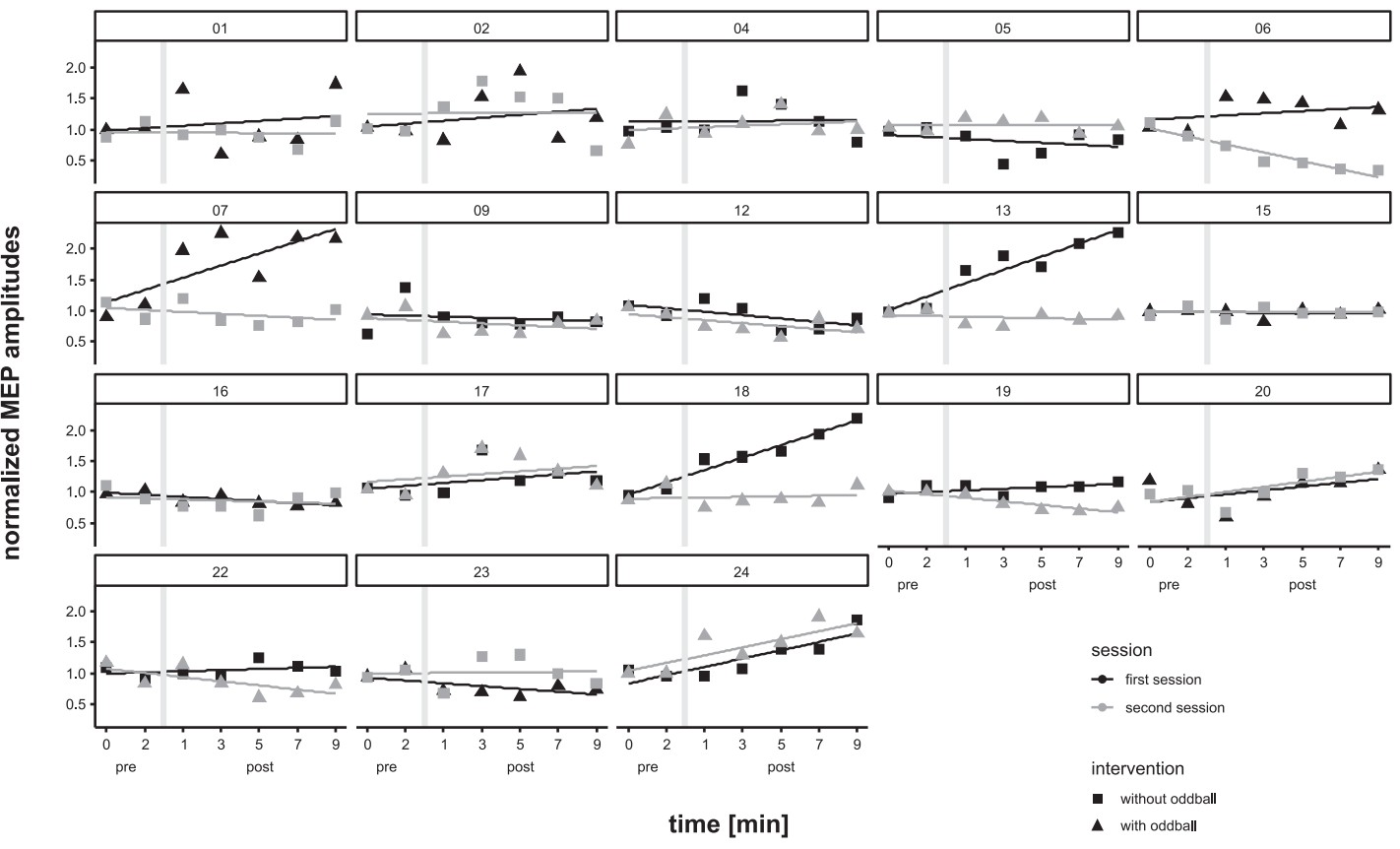

**Fig 3. Individual data.** Mean normalized MEP amplitudes are shown pre and post magnet exposure in 2 min intervals. The grey bars represent the exposure to the permanent magnet. Visual inspection yielded differences between the sessions only for the subjects 06, 07, 13 and 18, respectively.

again did not observe any modulatory effect of tSMS on excitability, the present study confirms our previous results.

Recent tSMS studies showed short-term modulatory effects up to 6min after application of tSMS for 10 [1] or 15 minutes [2]. In a recent study even long-lasting modulatory effects (at least 30min) on corticospinal excitability have been observed after tSMS applied for 30 minutes [6]. However, in our attempts to replicate tSMS modulatory effects we observed no significant difference in MEP amplitudes before and after 10 minutes of tSMS regardless of the application of the oddball task, which supports the results of our previous tSMS study [16].

A reason for the difference in our results compared to other tSMS studies might be high variability in corticospinal excitability and thus MEP amplitudes. In the present study we observed high intraindividual variabilities in baseline MEP amplitudes between session one and two, although RMTs and 1-mV-intensities correlated between the sessions. A comparable variability in baseline MEP amplitudes together with intersession reliability of RMTs has been reported recently in 27 subjects [24]. The observed variability in our study might mask weak modulatory effects caused by tSMS.

High variability in MEP amplitudes for single pulse TMS has been observed in other TMS studies. Several phenomena have been considered: spontaneous fluctuations in corticospinal and segmental motoneuron excitability levels [25], variation of synchronization and the number of activated motor units [26], and variability in corticospinal excitability and spinal

desynchronization [27]. In the latter study [27], another putative source of variability has been addressed, i.e. the stability of coil position. Variance of MEP amplitudes did not differ comparing navigated and non- navigated TMS in three consecutive measurements each. However, in another study the superiority of navigated TMS over non-navigated TMS has been demonstrated comparing MEP amplitudes [28]. The authors found higher MEP amplitudes combined with a lower variance when TMS was navigated, although MTs were similar with both methods. Since we used navigated TMS, in contrast to other published tSMS studies [e.g., 1, 2, 6], we cannot exclude that the observation of reduced MEP amplitudes in those studies might be due to the non-navigated TMS setup.

Another factor that could possibly influence the reliability of excitability measurements is the number of single TMS pulses averaged to a mean amplitude as dependent variable. Studies suggest the recording of at least 21 MEPs [29], or 26 for male and 30 for female subjects [30] for on optimal consistency in results. In our study we applied an average of 35 pulses (random jitter between 34 and 38) for each baseline measurement, which is above the recommended number. In other tSMS studies, baseline excitability measurements were based on 20 [1] or 30 [2] single TMS pulses. Another study indicates that the first 20 MEPs should not be used for excitability measurements, due to an initial transient-state of corticospinal excitability [31]. However, we observed no systematic difference between the two time points of each baseline measurement, although each time point included an average of only 18 pulses (random jitter between 16 and 19).

The choice of the exposed hemisphere might also contribute to the conflicting results. Whereas in the present study and in some of the previous tSMS experiments [2, 5, 16] the left hemisphere was exposed in right-handed subjects, in the first description of the effect [1] the right hemisphere was chosen. Unfortunately, no information about handedness of the subjects was provided in that study. One could speculate that the dominance of the hemisphere might have an influence on the intensity of putative inhibitory tSMS effects. Applying excitatory protocols, a more pronounced effect for the non-dominant hemisphere has been suggested with andoal tDCS [32, 33] and paired associative stimulation [34]. To our knowledge, there are no data with inhibitory protocols yet. Thus, the influence of hemispheric dominance on tSMS remains to be investigated.

The main independent variable in the present study was presence or absence of an acoustic oddball task during the application of tSMS. We observed no difference in MEP amplitudes in dependence of the oddball task. Moreover, in the subject´s first session, independent of the oddball task, MEPs tended to increase over time post stimulation, whereas there was no change in the subject´s second sessions. The tendency of MEP amplitudes to increase over time in the absence of any intervention was observed before [35–37]. However, in our data this effect is mainly driven by a subsample of four subjects (see Fig 3). Aside from the presence or absence of an acoustic oddball task the procedure in the two sessions of each subject was identical. So far, in all tSMS studies published the effects have not been repeated with the same settings. In addition, although the order of sham and real tSMS sessions was counterbalanced in those studies, their data have not been analyzed with respect to the factor order. Recently, for intermittent theta burst stimulation (iTBS) it was suggested that the modulatory effect might habituate if the same procedure was repeated [24]. In their study, the increase of corticospinal excitability following iTBS was present in a first session, but disappeared when a second, identical session was followed about one week apart. The authors concluded that there is a high interindividual variability as well as a low intraindividual reliability of iTBS effects, and that group results based on only one session should be interpreted with caution [24]. This notion does not only hold for iTBS. In a former study [23] we investigated the effects of transcranial direct current stimulation (tDCS) on visual cortex excitability. Subjects took part in

two sessions, which only differed after the tDCS intervention, but included an identical post-stimulation measurement. Only for cathodal tDCS a medium reliability was observed in the post-stimulation measurement, and overall a high variability in response to tDCS was found [23]. Therefore any study investigating the modulatory effect of a cortical stimulation including tSMS would be well advised to repeat the intervention in order to estimate the real effect.

In a study investigating the effects of three different non-invasive brain stimulation techniques on the excitability of the motor cortex, no distinct group effect was observed for either Paired Associative Stimulation, anodal tDCS or iTBS [38]. Instead, cluster analysis revealed a bimodal result pattern for all methods. However, less than half of the subjects responded as expected, and it was suggested that inter-individual variability has to be thoroughly addressed in the field of brain stimulation. Moreover, recently it was shown that cluster analyses based on MEPs might be sensitive for false positive results [37]. In that study, following four different classification techniques a significant number of subjects were classified as responders despite any intervention. Thus, the high variability might sometimes lead to false positive results if MEP amplitudes are used as the dependent variable. This, on the one hand, highlights the importance to unravel the underlying reasons for this variability [39]. On the other hand, since it was shown that neural activity can be regulated volitional [40], better methods have to be established to control for and to standardize brain activity before the application of brain stimulation techniques.

To conclude, we did not observe an inhibitory effect caused by tSMS, nor any influence of the auditory oddball task. However, these results do not exclude a tSMS effect in general. Further studies with adequate statistical power and within-subject replication would clarify this general question. Although there are first attempts for a potential treatment of neurological or psychiatric diseases using tSMS [41], the fundamental research of tSMS is still just at the beginning.

## Supporting information

**S1 File. Raw data.** File includes all data of the 18 subjects included in the analysis. (XLSX)

## Author Contributions

**Conceptualization:** Sabrina Lorenz, Birte Alex, Thomas Kammer.

**Data curation:** Sabrina Lorenz, Birte Alex, Thomas Kammer.

**Formal analysis:** Sabrina Lorenz.

**Investigation:** Birte Alex.

**Methodology:** Sabrina Lorenz, Thomas Kammer.

**Project administration:** Sabrina Lorenz, Thomas Kammer.

**Resources:** Thomas Kammer.

**Software:** Thomas Kammer.

**Supervision:** Thomas Kammer.

**Validation:** Sabrina Lorenz, Thomas Kammer.

**Writing – original draft:** Sabrina Lorenz, Birte Alex, Thomas Kammer.

**Writing – review & editing:** Sabrina Lorenz.

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
