## [Decision Letter · Decision Letter 0]

13 Feb 2020

PONE-D-19-34372

Transcranial static magnetic field stimulation does not reliably modulate motor cortex excitability

PLOS ONE

Dear Dr Lorenz,

Thank you for submitting your manuscript to PLOS ONE. I am very sorry for the longer review time, I had difficulties to find reviewers. Your paper was reviewed by two experts on the brain stimulation field, both of them had concerns with regard to the methodology and interpretation of the data. After careful consideration, we feel that it has merit but does not fully meet PLOS ONE’s publication criteria as it currently stands. Therefore, we invite you to submit a revised version of the manuscript that addresses all points raised during the review process.

We would appreciate receiving your revised manuscript by 14th of April, 2020. To enhance the reproducibility of your results, we recommend that if applicable you deposit your laboratory protocols in protocols.io, where a protocol can be assigned its own identifier (DOI) such that it can be cited independently in the future. For instructions see: http://journals.plos.org/plosone/s/submission-guidelines#loc-laboratory-protocols

We look forward to receiving your revised manuscript.

Kind regards,

Andrea Antal, PhD

Academic Editor

PLOS ONE

Journal Requirements:

Reviewers' comments:

Reviewer's Responses to Questions

**Comments to the Author**

1. Is the manuscript technically sound, and do the data support the conclusions?

Reviewer #1: Yes

Reviewer #2: No

2. Has the statistical analysis been performed appropriately and rigorously? 

Reviewer #1: No

Reviewer #2: Yes

3. Have the authors made all data underlying the findings in their manuscript fully available?

Reviewer #1: Yes

Reviewer #2: Yes

4. Is the manuscript presented in an intelligible fashion and written in standard English?

Reviewer #1: Yes

Reviewer #2: Yes

5. Review Comments to the Author

Reviewer #1: PONE-D-19-34372

Lorenz et al. investigated influence of tSMS on excitability of the primary motor cortex (M1). This is in essence a negative study pointing null effect of tSMS, regardless of inclusion of the oddball task; the only significant change was observed when the order of experimental sessions was taken into consideration. I believe null effects should be made public, as long as the methodology was rigorous and sound. In this regard, I have the following concerns.

1) Initially, the sample size had been calculated based on a certain medium effect size of f = 0.25 (page 5). What if this assumption was not the case? Did the authors calculate? If the actual effect size was smaller, more participants would be needed to detect a difference.

2) The significant effect of SESSION (first, second)*TIME interaction was observed in the second ANOVA with assumed sphericity. I understand this ANOVA was a (so to speak) post-hoc one; nevertheless, a three-way ANOVA could have been conducted first, using SESSION (with and without oddball), ORDER (first, second), and TIME as factors.

3) As a part of conclusion reads “absence of evidence is not evidence of absence (page 17).” I completely agree this statement, but just after that “Further studies with adequate statistical power” were claimed. We might not be able to say something about absence of evidence based on studies without adequate statistical power.

4) In Abstract, the authors argued habituation can be a key factor explaining the results, but this idea was not explicitly dealt in Discussion.

5) (Minor) sagittal “plain (page 6)” should be sagittal “plane.”

Reviewer #2: Review MS Nr. Pone-D-19-34372

This sham-controlled TMS-study was planned in order to examine the influence of cognitive activity (auditory oddball task) on tSMS- induced cortico-spinal excitability (CSE) changes. CSE was assessed via single pulse motor evoked potentials (MEP) in a pre- post experimental design. The results show that a period of 10 min tSMS (sham) did not modulate CSE, and there was no co-modulation from cognitive activity. The authors suggested that fluctuations in MEP amplitude could have masked possible weak modulatory effects. The study has a rationale and is clearly stated. However there are some points limiting its scientific value.

As described in the introduction, this study is a repetition an earlier TMS-study using 10 min tSMS (Kufner et al. 2017). Since after tSMS no effect was found in the earlier study, it is unlikely to find robust tSMS stimulation effects that are useful to clarify the additional influence of cognitive activity. In the introduction of this study I found no discussion how cognitive activity may interact with simultaneous tSMS with respect to pre-post-changes in motor cortical excitability and the formation neuroplasticity.

Although tSMS is described as a new simple form of inhibitory NIBS, and was highlighted a promising tool for brain stimulation, the mechanisms behind the neuromodulatory effects still remain unclear. Reorientation of diamagnetic anisotropic plasma membrane phospholipids (Rosen AD, 2003), coupling of mechanically-activated ion channels to ferromagnetic particles (Dobson et al. 1996) and cryptochromes (Landler & Keays 2018) are in discussion. Thus it should be stated somewhere that the fundamental research on tSMS is still at the beginning, and from this point it appears speculative to speak about treatment of neurological or psychiatric diseases.

Experimental part. The TMS-assessments for such a study are rather minimalistic. To gain more insight into the formation motor cortical plasticity and metaplasticity (tSMS + task) assessments of intracortical circuits by using by paired pulse TMS (SICI, SICF) showed advantageous (Dileone et. al. 2018). In the current study CSE was probed at a stimulation intensity to produce 1 mV MEPs. Here graded stimulation intensities resulting in MEP recruitment curves would provide information about the excitability changes of cortico-spinal circuits over a wider input-output range.

At least an exposure period (10 minutes tSMS) not appears optimal for such a study. Studies have shown that 10 minutes is the minimum, more stable and more long-lasting effect were obtained after periods of 20 and 30 min tSMS. From this point of view it is recommended to repeat the experiments with a longer stimulation period.

Minor points:

The title is no very representative for a study focusing on the influence of cognitive activity

Fig. 1 and Fig.2: the variations (standard error?) in the pre-assessment blocks are much smaller than in the post-assessment blocks. This is quite uncommon for TMS values, and needs explanation

6. PLOS authors have the option to publish the peer review history of their article (what does this mean?). If published, this will include your full peer review and any attached files.

Reviewer #1: No

Reviewer #2: No

---

## [Author Response · Author response to Decision Letter 0]

25 Mar 2020

Reviewer #1: PONE-D-19-34372

Lorenz et al. investigated influence of tSMS on excitability of the primary motor cortex (M1). This is in essence a negative study pointing null effect of tSMS, regardless of inclusion of the oddball task; the only significant change was observed when the order of experimental sessions was taken into consideration. I believe null effects should be made public, as long as the methodology was rigorous and sound. In this regard, I have the following concerns.

1) Initially, the sample size had been calculated based on a certain medium effect size of f = 0.25 (page 5). What if this assumption was not the case? Did the authors calculate? If the actual effect size was smaller, more participants would be needed to detect a difference.

Indeed, if the effect size would have been smaller, the number of participants needed would be higher. Unfortunately, in most of the previous reports on tSMS effects on motor cortex excitability (Oliviero et al., 2011; Silbert et al., 2013, Dileone et al., 2018) no effect size was reported. Thus, our power estimation was based on an assumed medium effect size of 0.25 (following Cohen in case of F-statistics). With an alpha error of 0.05 and a power of 0.8, for a 2 x 7 within factor design the calculated sample size is 18. We recruited 24 subjects and were lucky to be able to analyze exact 18 subjects. 

However, in most of the previous studies less subjects have been investigated (Oliviero et al., 2011: 11 subjects per experiment; Silbert et al., 2013: 10 subjects, Nojima et al., 2015: 10/10/10 per group; Dileone et al., 2018: 10/10/9/14(but 8 real 8 sham)/18(but 9 real 9 sham) for the 5 experiments, respectively). All cited studies demonstrated significant modulatory effects of tSMS. Therefore, it seems unlikely that our sample size was just too small to detect possible changes. 

Furthermore, Nojima et al., 2015 reported an effect size of η2= 0.163 for the Time*Group Interaction and of η2= 0.306 in the post hoc analysis for the significant suppression of MEP in the post-0 measurement, directly after tSMS exposure. Both values represent a high effect size, following Cohen (1988) who classified effect sizes above 0.14 as high. From that our assumption of a medium effect size is a conservative estimation.

2) The significant effect of SESSION (first, second)*TIME interaction was observed in the second ANOVA with assumed sphericity. I understand this ANOVA was a (so to speak) post-hoc one; nevertheless, a three-way ANOVA could have been conducted first, using SESSION (with and without oddball), ORDER (first, second), and TIME as factors.

Indeed, a three-way ANOVA including the factors SESSION (first, second, your “ORDER”), INTERVENTION (without and with oddball, your “SESSION”), and TIME can be conducted. See the analysis here:

Please note that in the manuscript, we used the factor named SESSION both for intervention type as well as for order. 

In this 3-way ANOVA the group of 18 subjects is splitted so that only 9 subjects are averaged for the factors SESSION and INTERVENTION, reducing the power compared to the approach presented in the manuscript. Furthermore, since our aim was to investigate the influence of the oddball-task on MEP Amplitude (i.e. SESSION (intervention)*TIME), we think that a 3-way ANOVA is not appropriate as first statistical attempt. The question of the influence of session order was a post-hoc question which should not be addressed in the first place. Therefore, we feel that the way we present the data is the appropriate one in a scientific sense. 

3) As a part of conclusion reads “absence of evidence is not evidence of absence (page 17).” I completely agree this statement, but just after that “Further studies with adequate statistical power” were claimed. We might not be able to say something about absence of evidence based on studies without adequate statistical power.

Thank you for this sophisticated hint. Indeed, without adequate statistical power neither evidence nor the absence of evidence can be stated. This does not touch the core problem, i.e. the generation of evidence of absence, which cannot be reached regardless the adequacy of power. 

We changed the passage in the conclusion section . It now reads: ”However, these results do not exclude a tSMS effect in general. Further studies with adequate statistical power and within-subject replication would clarify this general question.”

4) In Abstract, the authors argued habituation can be a key factor explaining the results, but this idea was not explicitly dealt in Discussion.

Thank you for indicating this discrepancy. It was not in our intention to postulate habituation as a key factor explaining our results. In the abstract, we only stated: “there might be a habituation effect to the whole procedure, resulting in less variability when subjects underwent the same experiment twice.”

 We tried to include the habituation argument in this passage of the discussion: “Recently, for intermittent theta burst stimulation (iTBS) it was suggested that the modulatory effect might habituate if the same procedure was repeated [24]. In their study, the increase of corticospinal excitability following iTBS was present in a first session, but disappeared when a second, identical session was followed about one week apart. The authors concluded that there is a high interindividual variability as well as a low intraindividual reliability of iTBS effects, and that group results based on only one session should be interpreted with caution [24].” (page 15f.). 

Since the difference between session 1 and 2 in our study is mainly driven by a subsample of 4 subjects (see Fig.3), a putative habituation on the whole procedure plays more a secondary role and was therefore not dealt with more attention. In fact, individual variability in MEP amplitudes seems to be smaller in the subject´s second sessions (which can be seen in the smaller standard errors in Fig. 2 as well), and we just argued that this might be due to a kind of habituation to the procedure. 

5) (Minor) sagittal “plain (page 6)” should be sagittal “plane.”

We corrected this spelling mistake.

Reviewer #2: Review MS Nr. Pone-D-19-34372

This sham-controlled TMS-study was planned in order to examine the influence of cognitive activity (auditory oddball task) on tSMS- induced cortico-spinal excitability (CSE) changes. CSE was assessed via single pulse motor evoked potentials (MEP) in a pre- post experimental design. The results show that a period of 10 min tSMS (sham) did not modulate CSE, and there was no co-modulation from cognitive activity. The authors suggested that fluctuations in MEP amplitude could have masked possible weak modulatory effects. The study has a rationale and is clearly stated. However there are some points limiting its scientific value.

As described in the introduction, this study is a repetition an earlier TMS-study using 10 min tSMS (Kufner et al. 2017). Since after tSMS no effect was found in the earlier study, it is unlikely to find robust tSMS stimulation effects that are useful to clarify the additional influence of cognitive activity. In the introduction of this study I found no discussion how cognitive activity may interact with simultaneous tSMS with respect to pre-post-changes in motor cortical excitability and the formation neuroplasticity.

Thank you for this request. We tried to explain the putative interaction between tSMS and cognitive activity introduced by the auditory oddball task. In the introduction, we wrote: 

"It was suggested that the use of an oddball task could offer an explanation for the difference in results [17]. Counting mentally could activate the motor cortex due to individual finger counting habits [18], and these habits might manipulate the structure of mental number representations [19]. It has also been observed that productive and perceptive linguistic tasks increase motor cortex excitability bilaterally, indicating that forming number words mentally while counting could cause activation in the hand representation of the motor cortex [20]." (page 3f.) 

We now added the following sentence: "Activation of the motor cortex without motor output might therefore interfere with neuroplastic modulation induced by static magnetic fields."

We think that this passage now clarifies both, the putative interaction of tSMS in the motor system and a cognitive activity involving motor cortex as well as the motivation for the study.

The reviewer is right with the notion that we did not find a tSMS effect in our previous study (Kufner et al. 2017). However, the problem here was that we only measured in combination with the auditory oddball task. Therefore, indeed, this cognitive task could have interacted with the modulatory influence of tSMS, destroying the putative tSMS effect. This exactly was postulated in a letter by Foffani et al. (2017). We cannot follow the argument of the reviewer that is unlikely to find robust tSMS effects in the light of our previous results (Kufner et al. 2017), since a replication of the study by Oliviero et al. (2011), without any additional cognitive task, should yield tSMS effects. 

Although tSMS is described as a new simple form of inhibitory NIBS, and was highlighted a promising tool for brain stimulation, the mechanisms behind the neuromodulatory effects still remain unclear. Reorientation of diamagnetic anisotropic plasma membrane phospholipids (Rosen AD, 2003), coupling of mechanically-activated ion channels to ferromagnetic particles (Dobson et al. 1996) and cryptochromes (Landler & Keays 2018) are in discussion. Thus it should be stated somewhere that the fundamental research on tSMS is still at the beginning, and from this point it appears speculative to speak about treatment of neurological or psychiatric diseases.

We agree that fundamental research on tSMS is at the beginning. Since other groups already published applications of tSMS in the context of neurological diseases, we stated in the introduction: " An effect of tSMS on cortical excitability comparable to those evoked by other noninvasive brain stimulation methods such as repetitive TMS [8-10] or transcranial direct current stimulation [11-13], but without the side effects [14, 15] would offer a wide range of possibilities for, i.e., treatment of neurological or psychiatric diseases." (page 3). However, in order to clarify our view on the early state of tSMS research, we added the following passage in the discussion: “Although there are first attempts for a potential treatment of neurological or psychiatric diseases using tSMS [41], the fundamental research of tSMS is still just at the beginning.”(page 17).

Experimental part. The TMS-assessments for such a study are rather minimalistic. To gain more insight into the formation motor cortical plasticity and metaplasticity (tSMS + task) assessments of intracortical circuits by using by paired pulse TMS (SICI, SICF) showed advantageous (Dileone et. al. 2018). In the current study CSE was probed at a stimulation intensity to produce 1 mV MEPs. Here graded stimulation intensities resulting in MEP recruitment curves would provide information about the excitability changes of cortico-spinal circuits over a wider input-output range.

We agree that besides MEP amplitude other parameters are suitable for measuring changes in cortico-spinal excitability. Since paired pulse TMS paradigms (SICI, ICF) show only moderate to poor test-retest-reliability whereas that for MEP amplitudes is better (i.e. Hermsen et al., 2016, J Neurol Sci 362), those measurements may not be that suitable to detect such small changes possibly caused by tSMS, although there was an effect reported by Dileone et al., 2018. However, all published data so far report a suppression effect of tSMS on MEP amplitude. Furthermore, the aim of the present study was explicitly to clarify the issue of putative interference of a cognitive task which was hypothesized to cancel the tSMS effect. Therefore, we chose MEP amplitude as the only dependent variable. 

At least an exposure period (10 minutes tSMS) not appears optimal for such a study. Studies have shown that 10 minutes is the minimum, more stable and more long-lasting effect were obtained after periods of 20 and 30 min tSMS. From this point of view it is recommended to repeat the experiments with a longer stimulation period.

We agree with the reviewer that a longer exposure of tSMS might increase the modulatory effect, as recent publications demonstrate. However, the aim of the study was to replicate the original finding, as well as to answer the question concerning the impact of the additional cognitive task, as raised by Foffani and Dileone (2017). Therefore, we did not prolong the exposure time. The main question of the study, namely whether the inhibitory effect of 10min tSMS reported by Oliviero et al., 2011 is abrogated if an acoustic oddball task was used during stimulation, would not be answered by using a prolonged stimulation time. 

Minor points:

The title is no very representative for a study focusing on the influence of cognitive activity

Thank you for this claim. Indeed, the title would not be appropriate for a study focusing on the influence of cognitive activity. However, in the present study we do not focus on the influence of cognitive activity. Instead, we focus on the reliability of putative tSMS effects on motor cortex excitability, trying to replicate former results and to investigate the reasons for replication failures. Therefore, we think that the title focusses on the main aspects of the manuscript.

Fig. 1 and Fig.2: the variations (standard error?) in the pre-assessment blocks are much smaller than in the post-assessment blocks. This is quite uncommon for TMS values, and needs explanation

We show normalized values. The normalization took place on the pre-assessment blocks. If we would average the two pre time points together to one single time point, no variation at all would be found since each and any subject, by definition, would have the value of 1. Only the fact that we kept the time resolution with two aggregation states in the pre-assessment blocks allowed us to include these two time points into the inference analysis.

---

## [Decision Letter · Decision Letter 1]

9 Apr 2020

PONE-D-19-34372R1

Transcranial static magnetic field stimulation does not reliably modulate motor cortex excitability

PLOS ONE

Dear Dr. Lorenz,

Thank you for submitting your manuscript to PLOS ONE. Your paper was reevaluated by the same Reviewers, they found that the manuscript has improved a lot, however, minor corrections/additions should be made before acceptance.  After careful consideration, we feel that it has merit but does not fully meet PLOS ONE’s publication criteria as it currently stands. Therefore, we invite you to submit a revised version of the manuscript that addresses all points raised during the review process.

We would appreciate receiving your revised manuscript by 10 of May, 2020. To enhance the reproducibility of your results, we recommend that if applicable you deposit your laboratory protocols in protocols.io, where a protocol can be assigned its own identifier (DOI) such that it can be cited independently in the future. For instructions see: http://journals.plos.org/plosone/s/submission-guidelines#loc-laboratory-protocols

We look forward to receiving your revised manuscript.

Kind regards,

Andrea Antal, PhD

Academic Editor

PLOS ONE

Reviewers' comments:

Reviewer's Responses to Questions

**Comments to the Author**

1. If the authors have adequately addressed your comments raised in a previous round of review and you feel that this manuscript is now acceptable for publication, you may indicate that here to bypass the “Comments to the Author” section, enter your conflict of interest statement in the “Confidential to Editor” section, and submit your "Accept" recommendation.

Reviewer #1: (No Response)

Reviewer #2: (No Response)

2. Is the manuscript technically sound, and do the data support the conclusions?

Reviewer #1: Yes

Reviewer #2: Yes

3. Has the statistical analysis been performed appropriately and rigorously? 

Reviewer #1: Yes

Reviewer #2: Yes

4. Have the authors made all data underlying the findings in their manuscript fully available?

Reviewer #1: Yes

Reviewer #2: Yes

5. Is the manuscript presented in an intelligible fashion and written in standard English?

Reviewer #1: Yes

Reviewer #2: Yes

6. Review Comments to the Author

Reviewer #1: I would like to thank the authors for the elaboration and revision. The responses to my previous comments are in principle satisfactory. Please include the ideas described in the comments in the main text.

More specifically,

1) Please report the effect size, as well as the p-values. Methods states that the power analysis was performed based on the f-value as an indicator for the effect size, whereas in the response to the comments eta-squared was presented. In the main text, please also be congruent.

2) Please describe explicitly why the authors considered the three-way ANOVA inappropriate in this study. Without a priori assumption it would be the first analysis.

Reviewer #2: The authors only made minor changes in the new version of the manuscript. Not all suggestions were replied, for example the use recruitment curves was recommended in the first review to test pre- post changes in motor cortex excitability (MCE)

The authors stated in their response that “aim of this study was to replicate the original finding…… as rised by Foffani and Dileone (2017)”. Thus the intention of this study was not to evaluate the effect of tSMS on MCE in relation to the previous research in this field. This would include longer stimulation times and additional TMS protocols. However the current title of their study suggests that tSMS per se produces no modulatory effect on MCE.

As solution to this problem a more result oriented title is proposed, for example: “Ten minutes of tSMS does not reliable modulate MCE”

7. PLOS authors have the option to publish the peer review history of their article (what does this mean?). If published, this will include your full peer review and any attached files.

Reviewer #1: No

Reviewer #2: No

---

## [Author Response · Author response to Decision Letter 1]

30 Apr 2020

Reviewer #1: I would like to thank the authors for the elaboration and revision. The responses to my previous comments are in principle satisfactory. Please include the ideas described in the comments in the main text.

More specifically,

1) Please report the effect size, as well as the p-values. Methods states that the power analysis was performed based on the f-value as an indicator for the effect size, whereas in the response to the comments eta-squared was presented. In the main text, please also be congruent.

Thank you for this hint. For the significant SESSION*TIME interaction in the analysis with respect to session order, we now added the missing effect size as η2 value (page 11). Furthermore, we rigorously checked all results of inference statistics reported with respect to completeness. 

Since our experiment was based on the first study by Oliviero et al., 2011 without reporting any effect sizes, in our former version of the manuscript we decided to estimate a medium effect size for power analysis, based on the f-value reported. However, due to the fact that in a more recent paper (Nojima et al., 2015) real effect sizes were reported and in line with our own calculation, we now decided to include a power analysis on the basis of the reported η2 values instead. We changed the passage in the methods section. It now reads: 

“Unfortunately, in the work of Oliviero et al., 2011 no effect size has been reported. Therefore, we based our power estimation (G*Power 3.1.7) on the effect size reported more recently (Nojima et al., 2015). In their analysis on MEP ratio for the left hand, they observed a significant Time by Group- interaction with an effect size of η2=0.163. Together with their sample size of n=20 (between- group) and an assumed α-error of 0.05, the power of their results was 0.24. Thus, using those values the calculated sample size for our experiment was 22. We decided to recruit 24 subjects to achieve appropriate power.”

2) Please describe explicitly why the authors considered the three-way ANOVA inappropriate in this study. Without a priori assumption it would be the first analysis.

We included the following passage in the results section:

“Furthermore, a friendly anonymous reviewer encouraged us to report a 3-way ANOVA including the factors SESSION (first/second), INTERVENTION (with/without oddball), and TIME (1-7). 

There was a significant effect for the factor SESSION (F(1,224)=12.35, p<0.001, η2= 0.05). However, there was no other significant effect for any factor (INTERVENTION: F(1,224)<0.001, p=0.99; TIME: F(6,224)=0.66, p=0.68) nor any statistically significant interaction. 

Please note that in this 3-way ANOVA the group of 18 subjects is splitted so that only 9 subjects are averaged for the factors SESSION and INTERVENTION. Furthermore, since our aim was to investigate the influence of the oddball-task on MEP Amplitude (i.e. INTERVENTION*TIME), the question of the influence of session order is a post-hoc question.” 

Reviewer #2: The authors only made minor changes in the new version of the manuscript. Not all suggestions were replied, for example the use recruitment curves was recommended in the first review to test pre- post changes in motor cortex excitability (MCE)

Thank you for indicating this. We are sorry for skipping a response to the topic recruitment curves. Indeed, recruitment curves seem to be more informative regarding modulation excitability in the motor system. However, it is more time-consuming to apply recruitment curves with respect to MEP recordings with a fixed intensity. Since we aimed to replicate several previous results, we restricted our dependent variable to MEP amplitudes.

The authors stated in their response that “aim of this study was to replicate the original finding…… as rised by Foffani and Dileone (2017)”. Thus the intention of this study was not to evaluate the effect of tSMS on MCE in relation to the previous research in this field. This would include longer stimulation times and additional TMS protocols. However the current title of their study suggests that tSMS per se produces no modulatory effect on MCE.

As solution to this problem a more result oriented title is proposed, for example: “Ten minutes of tSMS does not reliable modulate MCE”

Since the reviewer is right with his or her argument, we changed the title of the manuscript. It now reads “Ten minutes of transcranial static magnetic field stimulation does not reliably modulate motor cortex excitability”.

---

## [Decision Letter · Decision Letter 2]

11 May 2020

Ten minutes of transcranial static magnetic field stimulation does not reliably modulate motor cortex excitability

PONE-D-19-34372R2

Dear Dr. Lorenz,

We are pleased to inform you that your manuscript has been judged scientifically suitable for publication and will be formally accepted for publication once it complies with all outstanding technical requirements.

With kind regards,

Andrea Antal, PhD

Academic Editor

PLOS ONE

Additional Editor Comments (optional):

Reviewers' comments:

Reviewer's Responses to Questions

**Comments to the Author**

1. If the authors have adequately addressed your comments raised in a previous round of review and you feel that this manuscript is now acceptable for publication, you may indicate that here to bypass the “Comments to the Author” section, enter your conflict of interest statement in the “Confidential to Editor” section, and submit your "Accept" recommendation.

Reviewer #1: All comments have been addressed

Reviewer #2: All comments have been addressed

2. Is the manuscript technically sound, and do the data support the conclusions?

Reviewer #1: (No Response)

Reviewer #2: Yes

3. Has the statistical analysis been performed appropriately and rigorously? 

Reviewer #1: (No Response)

Reviewer #2: Yes

4. Have the authors made all data underlying the findings in their manuscript fully available?

Reviewer #1: (No Response)

Reviewer #2: Yes

5. Is the manuscript presented in an intelligible fashion and written in standard English?

Reviewer #1: (No Response)

Reviewer #2: Yes

6. Review Comments to the Author

Reviewer #1: (No Response)

Reviewer #2: The authors have nicely addressed all previous comments!

I have no further comments

7. PLOS authors have the option to publish the peer review history of their article (what does this mean?). If published, this will include your full peer review and any attached files.

Reviewer #1: No

Reviewer #2: No

---

## [Editor Report · Acceptance letter]

14 May 2020

PONE-D-19-34372R2 

Ten minutes of transcranial static magnetic field stimulation does not reliably modulate motor cortex excitability 

Dear Dr. Lorenz:

I am pleased to inform you that your manuscript has been deemed suitable for publication in PLOS ONE. Congratulations! Your manuscript is now with our production department. 

With kind regards,

on behalf of

Prof. Dr. Andrea Antal 

Academic Editor

PLOS ONE